# Streaming Robust Submodular Maximization: A Partitioned Thresholding Approach

**Slobodan Mitrović**[*]
EPFL

**Ilija Bogunović**[†]
EPFL

**Ashkan Norouzi-Fard**[‡]
EPFL

**Jakub Tarnawski**[§]
EPFL

**Volkan Cevher**[¶]
EPFL

## Abstract

We study the classical problem of maximizing a monotone submodular function subject to a cardinality constraint $k$, with two additional twists: (i) elements arrive in a streaming fashion, and (ii) $m$ items from the algorithm's memory are removed after the stream is finished. We develop a *robust* submodular algorithm STAR-T. It is based on a novel partitioning structure and an exponentially decreasing thresholding rule. STAR-T makes one pass over the data and retains a short but robust summary. We show that after the removal of any $m$ elements from the obtained summary, a simple greedy algorithm STAR-T-GREEDY that runs on the remaining elements achieves a constant-factor approximation guarantee. In two different data summarization tasks, we demonstrate that it matches or outperforms existing greedy and streaming methods, even if they are allowed the benefit of knowing the removed subset in advance.

## 1 Introduction

A central challenge in many large-scale machine learning tasks is data summarization – the extraction of a small representative subset out of a large dataset. Applications include image and document summarization [1, 2], influence maximization [3], facility location [4], exemplar-based clustering [5], recommender systems [6], and many more. Data summarization can often be formulated as the problem of maximizing a *submodular* set function subject to a cardinality constraint.

On small datasets, a popular algorithm is the simple greedy method [7], which produces solutions provably close to optimal. Unfortunately, it requires repeated access to all elements, which makes it infeasible for large-scale scenarios, where the entire dataset does not fit in the main memory. In this setting, streaming algorithms prove to be useful, as they make only a small number of passes over the data and use sublinear space.

In many settings, the extracted representative set is also required to be *robust*. That is, the objective value should degrade as little as possible when some elements of the set are removed. Such removals may arise for any number of reasons, such as failures of nodes in a network, or user preferences which the model failed to account for; they could even be adversarial in nature.

---

[*]e-mail: slobodan.mitrovic@epfl.ch

[†]e-mail: ilija.bogunovic@epfl.ch

[‡]e-mail: ashkan.norouzifard@epfl.ch

[§]e-mail: jakub.tarnawski@epfl.ch

[¶]e-mail: volkan.cevher@epfl.ch

A robustness requirement is especially challenging for large datasets, where it is prohibitively expensive to reoptimize over the entire data collection in order to find replacements for the removed elements. In some applications, where data is produced so rapidly that most of it is not being stored, such a search for replacements may not be possible at all.

These requirements lead to the following two-stage setting. In the first stage, we wish to solve the *robust streaming submodular maximization* problem – one of finding a small representative subset of elements that is robust against any possible removal of up to $m$ elements. In the second, *query* stage, after an arbitrary removal of $m$ elements from the summary obtained in the first stage, the goal is to return a representative subset, of size at most $k$, using only the precomputed summary rather than the entire dataset.

For example, (i) in *dominating set* problem (also studied under influence maximization) we want to efficiently (in a single pass) compute a compressed but robust set of influential users in a social network (whom we will present with free copies of a new product), (ii) in *personalized movie recommendation* we want to efficiently precompute a robust set of user-preferred movies. Once we discard those users who will not spread the word about our product, we should find a new set of influential users in the precomputed robust summary. Similarly, if some movies turn out not to be interesting for the user, we should still be able to provide good recommendations by only looking into our robust movie summary.

**Contributions.** In this paper, we propose a two-stage procedure for robust submodular maximization. For the first stage, we design a streaming algorithm which makes one pass over the data and finds a summary that is robust against removal of up to $m$ elements, while containing at most $O\left((m\log k + k)\log^2 k\right)$ elements.

In the second (query) stage, given any set of size $m$ that has been removed from the obtained summary, we use a simple greedy algorithm that runs on the remaining elements and produces a solution of size at most $k$ (without needing to access the entire dataset). We prove that this solution satisfies a constant-factor approximation guarantee.

Achieving this result requires novelty in the algorithm design as well as the analysis. Our streaming algorithm uses a structure where the constructed summary is arranged into partitions consisting of buckets whose sizes increase exponentially with the partition index. Moreover, buckets in different partitions are associated with greedy thresholds, which decrease exponentially with the partition index. Our analysis exploits and combines the properties of the described robust structure and decreasing greedy thresholding rule.

In addition to algorithmic and theoretical contributions, we also demonstrate in several practical scenarios that our procedure matches (and in some cases outperforms) the SIEVE-STREAMING algorithm [8] (see Section 5) – even though we allow the latter to know in advance which elements will be removed from the dataset.

## 2   Problem Statement

We consider a potentially large universe of elements $V$ of size $n$ equipped with a *normalized monotone submodular* set function $f : 2^V \to \mathbb{R}_{\geq 0}$ defined on $V$. We say that $f$ is *monotone* if for any two sets $X \subseteq Y \subseteq V$ we have $f(X) \leq f(Y)$. The set function $f$ is said to be *submodular* if for any two sets $X \subseteq Y \subseteq V$ and any element $e \in V \setminus Y$ it holds that

$$f(X \cup \{e\}) - f(X) \geq f(Y \cup \{e\}) - f(Y).$$

We use $f(Y \mid X)$ to denote the marginal gain in the function value due to adding the elements of set $Y$ to set $X$, i.e. $f(Y \mid X) := f(X \cup Y) - f(X)$. We say that $f$ is *normalized* if $f(\emptyset) = 0$.

The problem of maximizing a monotone submodular function subject to a cardinality constraint, i.e.,

$$\max_{Z \subseteq V, |Z| \leq k} f(Z), \tag{1}$$

has been studied extensively. It is well-known that a simple greedy algorithm (henceforth refered to as GREEDY) [7], which starts from an empty set and then iteratively adds the element with highest marginal gain, provides a $(1 - e^{-1})$-approximation. However, it requires repeated access to all elements of the dataset, which precludes it from use in large-scale machine learning applications.

We say that a set $S$ is *robust* for a parameter $m$ if, for any set $E \subseteq V$ such that $|E| \leq m$, there is a subset $Z \subseteq S \setminus E$ of size at most $k$ such that

$$f(Z) \geq cf(\text{OPT}(k, V \setminus E)),$$

where $c > 0$ is an approximation ratio. We use $\text{OPT}(k, V \setminus E)$ to denote the optimal subset of size $k$ of $V \setminus E$ (i.e., after the removal of elements in $E$):

$$\text{OPT}(k, V \setminus E) \in \underset{Z \subseteq V \setminus E, |Z| \leq k}{\text{argmax}} f(Z).$$

In this work, we are interested in solving a robust version of Problem (1) in the setting that consists of the following two stages: (i) *streaming* and (ii) *query* stage.

In the streaming stage, elements from the ground set $V$ arrive in a streaming fashion in an arbitrary order. Our goal is to design a one-pass streaming algorithm that has oracle access to $f$ and retains a small set $S$ of elements in memory. In addition, we want $S$ to be a robust summary, i.e., $S$ should both contain elements that maximize the objective value, and be robust against the removal of prespecified number of elements $m$. In the query stage, after any set $E$ of size at most $m$ is removed from $V$, the goal is to return a set $Z \subseteq S \setminus E$ of size at most $k$ such that $f(Z)$ is maximized.

**Related work.** A robust, non-streaming version of Problem (1) was first introduced in [9]. In that setting, the algorithm must output a set $Z$ of size $k$ which maximizes the smallest objective value guaranteed to be obtained after a set of size $m$ is removed, that is,

$$\max_{Z \subseteq V, |Z| \leq k} \min_{E \subseteq Z, |E| \leq m} f(Z \setminus E).$$

The work [10] provides the first constant (0.387) factor approximation result to this problem, valid for $m = o(\sqrt{k})$. Their solution consists of buckets of size $O(m^2 \log k)$ that are constructed greedily, one after another. Recently, in [11], a centralized algorithm PRO has been proposed that achieves the same approximation result and allows for a greater robustness $m = o(k)$. PRO constructs a set that is arranged into partitions consisting of buckets whose sizes increase exponentially with the partition index. In this work, we use a similar structure for the robust set but, instead of filling the buckets greedily one after another, we place an element in the first bucket for which the gain of adding the element is above the corresponding threshold. Moreover, we introduce a novel analysis that allows us to be robust to any number of removals $m$ as long as we are allowed to use $O(m \log^2 k)$ memory.

Recently, submodular streaming algorithms (e.g. [5], [12] and [13]) have become a prominent option for scaling submodular optimization to large-scale machine learning applications. A popular submodular streaming algorithm SIEVE-STREAMING [8] solves Problem (1) by performing one pass over the data, and achieves a $(0.5 - \epsilon)$-approximation while storing at most $O\left(\frac{k \log k}{\epsilon}\right)$ elements.

Our algorithm extends the algorithmic ideas of SIEVE-STREAMING, such as greedy thresholding, to the robust setting. In particular, we introduce a new exponentially decreasing thresholding scheme that, together with an innovative analysis, allows us to obtain a constant-factor approximation for the robust streaming problem.

Recently, robust versions of submodular maximization have been considered in the problems of influence maximization (e.g, [3], [14]) and budget allocation ([15]). Increased interest in interactive machine learning methods has also led to the development of interactive and adaptive submodular optimization (see e.g. [16], [17]). Our procedure also contains the interactive component, as we can compute the robust summary only once and then provide different sub-summaries that correspond to multiple different removals (see Section 5.2).

Independently and concurrently with our work, [18] gave a streaming algorithm for robust submodular maximization under the cardinality constraint. Their approach provides a $1/2 - \varepsilon$ approximation guarantee. However, their algorithm uses $O(mk \log k/\varepsilon)$ memory. While the memory requirement of their method increases linearly with $k$, in the case of our algorithm this dependence is logarithmic.

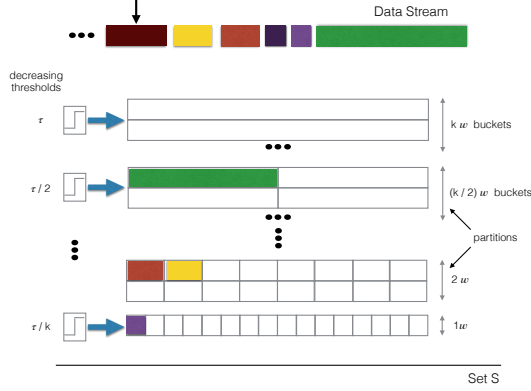

Figure 1: Illustration of the set $S$ returned by STAR-T. It consists of $\lceil \log k \rceil + 1$ partitions such that each partition $i$ contains $w\lceil k/2^i \rceil$ buckets of size $2^i$ (up to rounding). Moreover, each partition $i$ has its corresponding threshold $\tau/2^i$.

## 3 A Robust Two-stage Procedure

Our approach consists of the streaming Algorithm 1, which we call Streaming Robust submodular algorithm with Partitioned Thresholding (STAR-T). This algorithm is used in the streaming stage, while Algorithm 2, which we call STAR-T-GREEDY, is used in the query stage.

As the input, STAR-T requires a non-negative monotone submodular function $f$, cardinality constraint $k$, robustness parameter $m$ and thresholding parameter $\tau$. The parameter $\tau$ is an $\alpha$-approximation to $f(\text{OPT}(k, V \setminus E))$, for some $\alpha \in (0, 1]$ to be specified later. Hence, it depends on $f(\text{OPT}(k, V \setminus E))$, which is not known a priori. For the sake of clarity, we present the algorithm as if $f(\text{OPT}(k, V \setminus E))$ were known, and in Section 4.1 we show how $f(\text{OPT}(k, V \setminus E))$ can be approximated. The algorithm makes one pass over the data and outputs a set of elements $S$ that is later used in the query stage in STAR-T-GREEDY.

The set $S$ (see Figure 1 for an illustration) is divided into $\lceil \log k \rceil + 1$ partitions, where every partition $i \in \{0, \ldots, \lceil \log k \rceil\}$ consists of $w\lceil k/2^i \rceil$ buckets $B_{i,j}, j \in \{1, \ldots, w\lceil k/2^i \rceil\}$. Here, $w \in \mathbb{N}_+$ is a memory parameter that depends on $m$; we use $w \geq \left\lceil \frac{4\lceil \log k \rceil m}{k} \right\rceil$ in our asymptotic theory, while our numerical results show that $w = 1$ works well in practice. Every bucket $B_{i,j}$ stores at most $\min\{k, 2^i\}$ elements. If $|B_{i,j}| = \min\{2^i, k\}$, then we say that $B_{i,j}$ is *full*.

Every partition has a corresponding threshold that is exponentially decreasing with the partition index $i$ as $\tau/2^i$. For example, the buckets in the first partition will only store elements that have marginal value at least $\tau$. Every element $e \in V$ arriving on the stream is assigned to the first non-full bucket $B_{i,j}$ for which the marginal value $f(e \mid B_{i,j})$ is at least $\tau/2^i$. If there is no such bucket, the element will not be stored. Hence, the buckets are disjoint sets that in the end (after one pass over the data) can have a smaller number of elements than specified by their corresponding cardinality constraints, and some of them might even be empty. The set $S$ returned by STAR-T is the union of all the buckets.

In the second stage, STAR-T-GREEDY receives as input the set $S$ constructed in the streaming stage, a set $E \subset S$ that we think of as removed elements, and the cardinality constraint $k$. The algorithm then returns a set $Z$, of size at most $k$, that is obtained by running the simple greedy algorithm GREEDY on the set $S \setminus E$. Note that STAR-T-GREEDY can be invoked for different sets $E$.

## 4 Theoretical Bounds

In this section we discuss our main theoretical results. We initially assume that the value $f(\text{OPT}(k, V \setminus E))$ is known; later, in Section 4.1, we remove this assumption. The more detailed versions of our proofs are given in the supplementary material. We begin by stating the main result.

---

**Algorithm 1** STreAming Robust - Thresholding submodular algorithm (STAR-T)

---

**Input:** Set $V, k, \tau, w \in \mathbb{N}_+$
1: $B_{i,j} \leftarrow \emptyset$  for all $0 \leq i \leq \lceil \log k \rceil$ and $1 \leq j \leq w \lceil k/2^i \rceil$
2: **for each** element $e$ in the stream **do**
3:    **for** $i \leftarrow 0$ **to** $\lceil \log k \rceil$ **do**                     ▷ loop over partitions
4:       **for** $j \leftarrow 1$ **to** $w \lceil k/2^i \rceil$ **do**                ▷ loop over buckets
5:          **if** $|B_{i,j}| < \min\{2^i, k\}$ **and** $f(e \mid B_{i,j}) \geq \tau / \min\{2^i, k\}$ **then**
6:             $B_{i,j} \leftarrow B_{i,j} \cup \{e\}$
7:             **break:** proceed to the next element in the stream
8: $S \leftarrow \bigcup_{i,j} B_{i,j}$
9: **return** $S$

---

**Algorithm 2** STAR-T- GREEDY

---

**Input:** Set $S$, query set $E$ and $k$
1: $Z \leftarrow \text{GREEDY}(k, S \setminus E)$
2: **return** $Z$

---

**Theorem 4.1** *Let $f$ be a normalized monotone submodular function defined over the ground set $V$. Given a cardinality constraint $k$ and parameter $m$, for a setting of parameters $w \geq \left\lceil \frac{4\lceil \log k \rceil m}{k} \right\rceil$ and*

$$\tau = \frac{1}{2 + \frac{(1-e^{-1})}{(1-e^{-1/3})}\left(1 - \frac{1}{\lceil \log k \rceil}\right)} f(\text{OPT}(k, V \setminus E)),$$

STAR-T *performs a single pass over the data set and constructs a set $S$ of size at most $O((k + m \log k) \log k)$ elements.*

*For such a set $S$ and any set $E \subseteq V$ such that $|E| \leq m$,* STAR-T-GREEDY *yields a set $Z \subseteq S \setminus E$ of size at most $k$ with*

$$f(Z) \geq c \cdot f(\text{OPT}(k, V \setminus E)),$$

*for $c = 0.149 \left(1 - \frac{1}{\lceil \log k \rceil}\right)$. Therefore, as $k \to \infty$, the value of $c$ approaches $0.149$.*

**Proof sketch.**    We first consider the case when there is a partition $i^\star$ in $S$ such that at least half of its buckets are full. We show that there is at least one full bucket $B_{i^\star,j}$ such that $f(B_{i^\star,j} \setminus E)$ is only a constant factor smaller than $f(\text{OPT}(k, V \setminus E))$, as long as the threshold $\tau$ is set close to $f(\text{OPT}(k, V \setminus E))$. We make this statement precise in the following lemma:

**Lemma 4.2** *If there exists a partition in $S$ such that at least half of its buckets are full, then for the set $Z$ produced by* STAR-T-GREEDY *we have*

$$f(Z) \geq \left(1 - e^{-1}\right)\left(1 - \frac{4m}{wk}\right)\tau. \tag{2}$$

To prove this lemma, we first observe that from the properties of GREEDY it follows that

$$f(Z) = f(\text{GREEDY}(k, S \setminus E)) \geq \left(1 - e^{-1}\right) f(B_{i^\star,j} \setminus E).$$

Now it remains to show that $f(B_{i^\star,j} \setminus E)$ is close to $\tau$. We observe that for any full bucket $B_{i^\star,j}$, we have $|B_{i^\star,j}| = \min\{2^i, k\}$, so its objective value $f(B_{i^\star,j})$ is at least $\tau$ (every element added to this bucket increases its objective value by at least $\tau / \min\{2^i, k\}$). On average, $|B_{i^\star,j} \cap E|$ is relatively small, and hence we can show that there exists some full bucket $B_{i^\star,j}$ such that $f(B_{i^\star,j} \setminus E)$ is close to $f(B_{i^\star,j})$.

Next, we consider the other case, i.e., when for every partition, more than half of its buckets are not full after the execution of STAR-T. For every partition $i$, we let $B_i$ denote a bucket that is not fully populated and for which $|B_i \cap E|$ is minimized over all the buckets of that partition. Then, we look at such a bucket in the last partition: $B_{\lceil \log k \rceil}$.

We provide two lemmas that depend on $f(B_{\lceil \log k \rceil})$. If $\tau$ is set to be small compared to $f(\text{OPT}(k, V \setminus E))$:

- Lemma 4.3 shows that if $f(B_{\lceil \log k \rceil})$ is close to $f(\mathrm{OPT}(k, V \setminus E))$, then our solution is within a constant factor of $f(\mathrm{OPT}(k, V \setminus E))$;
- Lemma 4.4 shows that if $f(B_{\lceil \log k \rceil})$ is small compared to $f(\mathrm{OPT}(k, V \setminus E))$, then our solution is again within a constant factor of $f(\mathrm{OPT}(k, V \setminus E))$.

**Lemma 4.3** *If there does not exist a partition of $S$ such that at least half of its buckets are full, then for the set $Z$ produced by* STAR-T-GREEDY *we have*

$$f(Z) \geq \left(1 - e^{-1/3}\right) \left( f\left(B_{\lceil \log k \rceil}\right) - \frac{4m}{wk}\tau \right),$$

*where $B_{\lceil \log k \rceil}$ is a not-fully-populated bucket in the last partition that minimizes $\left| B_{\lceil \log k \rceil} \cap E \right|$ and $|E| \leq m$.*

Using standard properties of submodular functions and the GREEDY algorithm we can show that

$$f(Z) = f(\mathrm{GREEDY}(k, S \setminus E)) \geq \left(1 - e^{-1/3}\right) \left( f\left(B_{\lceil \log k \rceil}\right) - \frac{4m}{wk}\tau \right).$$

The complete proof of this result can be found in Lemma B.2, in the supplementary material.

**Lemma 4.4** *If there does not exist a partition of $S$ such that at least half of its buckets are full, then for the set $Z$ produced by* STAR-T-GREEDY,

$$f(Z) \geq (1 - e^{-1})\big(f(OPT(k, V \setminus E)) - f(B_{\lceil \log k \rceil}) - \tau\big),$$

*where $B_{\lceil \log k \rceil}$ is any not-fully-populated bucket in the last partition.*

To prove this lemma, we look at two sets $X$ and $Y$, where $Y$ contains all the elements from $\mathrm{OPT}(k, V \setminus E)$ that are placed in the buckets that precede bucket $B_{\lceil \log k \rceil}$ in $S$, and set $X := \mathrm{OPT}(k, V \setminus E) \setminus Y$. By monotonicity and submodularity of $f$, we bound $f(Y)$ by:

$$f(Y) \geq f(\mathrm{OPT}(k, V \setminus E)) - f(X) \geq f(\mathrm{OPT}(k, V \setminus E)) - f\left(B_{\lceil \log k \rceil}\right) - \sum_{e \in X} f\left(e \mid B_{\lceil \log k \rceil}\right).$$

To bound the sum on the right hand side we use that for every $e \in X$ we have $f\left(e \mid B_{\lceil \log k \rceil}\right) < \frac{\tau}{k}$, which holds due to the fact that $B_{\lceil \log k \rceil}$ is a bucket in the last partition and is not fully populated.

We conclude the proof by showing that $f(Z) = f(\mathrm{GREEDY}(k, S \setminus E)) \geq \left(1 - e^{-1}\right) f(Y)$.

Equipped with the above results, we proceed to prove our main result.

*Proof of Theorem 4.1.* First, we prove the bound on the size of $S$:

$$|S| = \sum_{i=0}^{\lceil \log k \rceil} w \lceil k/2^i \rceil \min\{2^i, k\} \leq \sum_{i=0}^{\lceil \log k \rceil} w(k/2^i + 1)2^i \leq (\log k + 5)wk. \tag{3}$$

By setting $w \geq \left\lceil \frac{4\lceil \log k \rceil m}{k} \right\rceil$ we obtain $S = O((k + m \log k) \log k)$.

Next, we show the approximation guarantee. We first define $\gamma := \frac{4m}{wk}$, $\alpha_1 := \left(1 - e^{-1/3}\right)$, and $\alpha_2 := \left(1 - e^{-1}\right)$. Lemma 4.3 and 4.4 provide two bounds on $f(Z)$, one increasing and one decreasing in $f(B_{\lceil \log k \rceil})$. By balancing out the two bounds, we derive

$$f(Z) \geq \left( \frac{\alpha_1 \alpha_2}{\alpha_1 + \alpha_2} \right) (f(\mathrm{OPT}(k, V \setminus E)) - (1 + \gamma)\tau), \tag{4}$$

with equality for $f(B_{\lceil \log k \rceil}) = \frac{\alpha_2 f(\mathrm{OPT}(k, V \setminus E)) - (\alpha_2 - \gamma \alpha_1)\tau}{\alpha_2 + \alpha_1}$.

Next, as $\gamma \geq 0$, we can observe that Eq. (4) is decreasing, while the bound on $f(Z)$ given by Lemma 4.2 is increasing in $\tau$ for $\gamma < 1$. Hence, by balancing out the two inequalities, we obtain our final bound

$$f(Z) \geq \frac{1}{\frac{2}{\alpha_2(1-\gamma)} + \frac{1}{\alpha_1}} f(\mathrm{OPT}(k, V \setminus E)). \tag{5}$$

For $w \geq \left\lceil \frac{4\lceil \log k \rceil m}{k} \right\rceil$ we have $\gamma \leq 1/\lceil \log k \rceil$, and hence, by substituting $\alpha_1$ and $\alpha_2$ in Eq. (5), we prove our main result:

$$
\begin{aligned}
f(Z) &\geq \frac{\left(1 - e^{-1/3}\right)\left(1 - e^{-1}\right)\left(1 - \frac{1}{\lceil \log k \rceil}\right)}{2\left(1 - e^{-1/3}\right) + \left(1 - e^{-1}\right)} f(\mathrm{OPT}(k, V \setminus E)) \\
&\geq 0.149 \left(1 - \frac{1}{\lceil \log k \rceil}\right) f(\mathrm{OPT}(k, V \setminus E)).
\end{aligned}
$$

$\square$

## 4.1 Algorithm without access to $f(\mathrm{OPT}(k, V \setminus E))$

Algorithm STAR-T requires in its input a parameter $\tau$ which is a function of an unknown value $f(\mathrm{OPT}(k, V \setminus E))$. To deal with this shortcoming, we show how to extend the idea of [8] of maintaining multiple parallel instances of our algorithm in order to approximate $f(\mathrm{OPT}(k, V \setminus E))$. For a given constant $\epsilon > 0$, this approach increases the space by a factor of $\log_{1+\epsilon} k$ and provides a $(1 + \epsilon)$-approximation compared to the value obtained in Theorem 4.1. More precisely, we prove the following theorem.

**Theorem 4.5** *For any given constant $\epsilon > 0$ there exists a parallel variant of* STAR-T *that makes one pass over the stream and outputs a collection of sets $\mathcal{S}$ of total size $O\left((k + m \log k) \log k \log_{1+\epsilon} k\right)$ with the following property: There exists a set $S \in \mathcal{S}$ such that applying* STAR-T-GREEDY *on $S$ yields a set $Z \subseteq S \setminus E$ of size at most $k$ with*

$$
f(Z) \geq \frac{0.149}{1 + \epsilon}\left(1 - \frac{1}{\lceil \log k \rceil}\right) f(\mathrm{OPT}(k, V \setminus E)).
$$

The proof of this theorem, along with a description of the corresponding algorithm, is provided in Appendix E.

## 5 Experiments

In this section, we numerically validate the claims outlined in the previous section. Namely, we test the robustness and compare the performance of our algorithm against the SIEVE-STREAMING algorithm that knows in advance which elements will be removed. We demonstrate improved or matching performance in two different data summarization applications: (i) the dominating set problem, and (ii) personalized movie recommendation. We illustrate how a single robust summary can be used to regenerate recommendations corresponding to multiple different removals.

### 5.1 Dominating Set

In the dominating set problem, given a graph $G = (V, M)$, where $V$ represents the set of nodes and $M$ stands for edges, the objective function is given by $f(Z) = |\mathcal{N}(Z) \cup Z|$, where $\mathcal{N}(Z)$ denotes the neighborhood of $Z$ (all nodes adjacent to any node of $Z$). This objective function is monotone and submodular.

We consider two datasets: (i) ego-Twitter [19], consisting of 973 social circles from Twitter, which form a directed graph with 81306 nodes and 1768149 edges; (ii) Amazon product co-purchasing network [20]: a directed graph with 317914 nodes and 1745870 edges.

Given the dominating set objective function, we run STAR-T to obtain the robust summary $S$. Then we compare the performance of STAR-T-GREEDY, which runs on $S$, against the performance of SIEVE-STREAMING, which we allow to know in advance which elements will be removed. We also compare against a method that chooses the same number of elements as STAR-T, but does so uniformly at random from the set of all elements that will not be removed ($V \setminus E$); we refer to it as RANDOM. Finally, we also demonstrate the peformance of STAR-T-SIEVE, a variant of our algorithm that uses the same robust summary $S$, but instead of running GREEDY in the second stage, it runs SIEVE-STREAMING on $S \setminus E$.

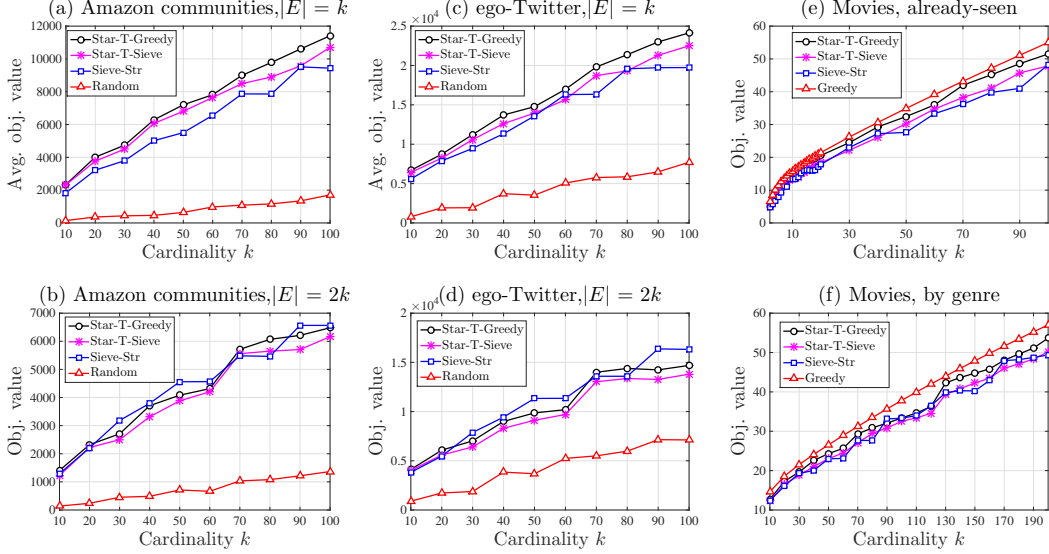

Figure 2: Numerical comparisons of the algorithms STAR-T-GREEDY, STAR-T-SIEVE and SIEVE-STREAMING.

Figures 2(a,c) show the objective value after the random removal of $k$ elements from the set $S$, for different values of $k$. Note that $E$ is sampled as a subset of the summary of *our* algorithm, which hurts the performance of our algorithm more than the baselines. The reported numbers are averaged over 100 iterations. STAR-T-GREEDY, STAR-T-SIEVE and SIEVE-STREAMING perform comparably (STAR-T-GREEDY slightly outperforms the other two), while RANDOM is significantly worse.

In Figures 2(b,d) we plot the objective value for different values of $k$ after the removal of $2k$ elements from the set $S$, chosen greedily (i.e., by iteratively removing the element that reduces the objective value the most). Again, STAR-T-GREEDY, STAR-T-SIEVE and SIEVE-STREAMING perform comparably, but this time SIEVE-STREAMING slightly outperforms the other two for some values of $k$. We observe that even when we remove more than $k$ elements from $S$, the performance of our algorithm is still comparable to the performance of SIEVE-STREAMING (which knows in advance which elements will be removed). We provide additional results in the supplementary material.

## 5.2 Interactive Personalized Movie Recommendation

The next application we consider is personalized movie recommendation. We use the MovieLens 1M database [21], which contains 1000209 ratings for 3900 movies by 6040 users. Based on these ratings, we obtain feature vectors for each movie and each user by using standard low-rank matrix completion techniques [22]; we choose the number of features to be 30.

For a user $u$, we use the following monotone submodular function to recommend a set of movies $Z$:

$$f_u(Z) = (1 - \alpha) \cdot \sum_{z \in Z} \langle v_u, v_z \rangle + \alpha \cdot \sum_{m \in M} \max_{z \in Z} \langle v_m, v_z \rangle.$$

The first term aggregates the predicted scores of the chosen movies $z \in Z$ for the user $u$ (here $v_u$ and $v_z$ are non-normalized feature vectors of user $u$ and movie $z$, respectively). The second term corresponds to a facility-location objective that measures how well the set $Z$ covers the set of all movies $M$ [4]. Finally, $\alpha$ is a user-dependent parameter that specifies the importance of global movie coverage versus high scores of individual movies.

Here, the robust setting arises naturally since we do not have complete information about the user: when shown a collection of top movies, it will likely turn out that they have watched (but not rated) many of them, rendering these recommendations moot. In such an interactive setting, the user may also require (or exclude) movies of a specific genre, or similar to some favorite movie.

We compare the performance of our algorithms STAR-T-GREEDY and STAR-T-SIEVE in such scenarios against two baselines: GREEDY and SIEVE-STREAMING (both being run on the set $V \setminus E$, i.e., knowing the removed elements in advance). Note that in this case we are able to afford running

GREEDY, which may be infeasible when working with larger datasets. Below we discuss two concrete practical scenarios featured in our experiments.

**Movies by genre.** After we have built our summary $S$, the user decides to watch a drama today; we retrieve only movies of this genre from $S$. This corresponds to removing $59\%$ of the universe $V$. In Figure 2(f) we report the quality of our output compared to the baselines (for user ID $445$ and $\alpha = 0.95$) for different values of $k$. The performance of STAR-T-GREEDY is within several percent of the performance of GREEDY (which we can consider as a tractable optimum), and the two sieve-based methods STAR-T-SIEVE and SIEVE-STREAMING display similar objective values.

**Already-seen movies.** We randomly sample a set $E$ of movies already watched by the user (500 out of all 3900 movies). To obtain a realistic subset, each movie is sampled proportionally to its popularity (number of ratings). Figure 2(e) shows the performance of our algorithm faced with the removal of $E$ (user ID $= 445$, $\alpha = 0.9$) for a range of settings of $k$. Again, our algorithm is able to almost match the objective values of GREEDY (which is aware of $E$ in advance).

Recall that we are able to use the same precomputed summary $S$ for different removed sets $E$. This summary was built for parameter $w = 1$, which theoretically allows for up to $k$ removals. However, despite having $|E| \gg k$ in the above scenarios, our performance remains robust; this indicates that our method is more resilient in practice than what the proved bound alone would guarantee.

## 6    Conclusion

We have presented a new robust submodular streaming algorithm STAR-T based on a novel partitioning structure and an exponentially decreasing thresholding rule. It makes one pass over the data and retains a set of size $O\left((k + m \log k) \log^2 k\right)$. We have further shown that after the removal of any $m$ elements, a simple greedy algorithm that runs on the obtained set achieves a constant-factor approximation guarantee for robust submodular function maximization. In addition, we have presented two numerical studies where our method compares favorably against the SIEVE-STREAMING algorithm that knows in advance which elements will be removed.

**Acknowledgment.** IB and VC's work was supported in part by the European Research Council (ERC) under the European Union's Horizon 2020 research and innovation program (grant agreement number 725594), in part by the Swiss National Science Foundation (SNF), project 407540_167319/1, in part by the NCCR MARVEL, funded by the Swiss National Science Foundation, in part by Hasler Foundation Switzerland under grant agreement number 16066 and in part by Office of Naval Research (ONR) under grant agreement number N00014-16-R-BA01. JT's work was supported by ERC Starting Grant 335288-OptApprox.

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
