[Supplementary Material · supplementary-material.pdf]

# Supplementary Material

This section presents the complete proofs of lemmas presented in the article.

## A   Detailed Proof of Lemma 4.2

**Lemma 4.2** *If there exists a partition in $S$ such that at least half of its buckets are full, then for the set $Z$ produced by* STAR-T-GREEDY *we have*

$$f(Z) \geq \left(1 - e^{-1}\right)\left(1 - \frac{4m}{wk}\right)\tau. \tag{2}$$

*Proof.* Let $i^\star$ be a partition such that half of its buckets are full. Let $B_{i^\star,j}$ be a full bucket that minimizes $|B_{i^\star,j} \cap E|$. In STAR-T, every partition contains $w\lceil k/2^i \rceil$ buckets. Hence, the number of full buckets in partition $i^\star$ is at least $wk/2^{i^\star+1}$. That further implies

$$|B_{i^\star,j} \cap E| \leq \frac{2^{i^\star+1}m}{wk}. \tag{6}$$

Taking into account that $B_{i^\star,j}$ is a full bucket, we conclude

$$|B_{i^\star,j} \setminus E| \geq |B_{i^\star,j}| - \frac{2^{i^\star+1}m}{wk}. \tag{7}$$

From the property of our Algorithm (line 5) every element added to $B_{i^\star,j}$ increased the utility of this bucket by at least $\tau/2^{i^\star}$. Combining this with the fact that $B_{i^\star,j}$ is full, we conclude that the gain of every element in this bucket is at least $\tau/|B_{i^\star,j}|$. Therefore, from Eq. (7) it follows:

$$f\left(B_{i^\star,j} \setminus E\right) \geq \left(|B_{i^\star,j}| - \frac{2^{i^\star+1}m}{wk}\right)\frac{\tau}{|B_{i^\star,j}|} = \tau\left(1 - \frac{2^{i^\star+1}m}{|B_{i^\star,j}|\,wk}\right). \tag{8}$$

Taking into account that $2^{i^\star+1} \leq 4\,|B_{i^\star,j}|$ this further reduces to

$$f\left(B_{i^\star,j} \setminus E\right) \geq \tau\left(1 - \frac{4m}{wk}\right). \tag{9}$$

Finally,

$$
\begin{aligned}
f(Z) = f(\text{GREEDY}(k, S \setminus E)) &\geq (1 - e^{-1})f(\text{OPT}(k, S \setminus E)) \\
&\geq \left(1 - e^{-1}\right)f(\text{OPT}(k, B_{i^\star,j} \setminus E)) \tag{10} \\
&= \left(1 - e^{-1}\right)f\left(B_{i^\star,j} \setminus E\right) \tag{11} \\
&\geq \left(1 - e^{-1}\right)\left(1 - \frac{4m}{wk}\right)\tau, \tag{12}
\end{aligned}
$$

where Eq. (10) follows from $(B_{i^\star,j} \setminus E) \subseteq (S \setminus E)$, Eq. (11) follows from the fact that $|B_{i^\star,j}| \leq k$, and Eq. (12) follows from Eq. (9). □

## B   Detailed Proof of Lemma 4.3

We start by studying some properties of $E$ that we use in the proof of Lemma 4.3.

**Lemma B.1** *Let $B_i$ be a bucket in partition $i > 0$, and let $E_i := B_i \cap E$ denote the elements that are removed from this bucket. Given a bucket $B_{i-1}$ from the previous partition such that $|B_{i-1}| < 2^{i-1}$ (i.e. $B_{i-1}$ is not fully populated), the loss in the bucket $B_i$ due to the removals is at most*

$$f\left(E_i \mid B_{i-1}\right) < \frac{\tau}{2^{i-1}}|E_i|.$$

*Proof.* First, we can bound $f(E_i \mid B_{i-1})$ as follows

$$f(E_i \mid B_{i-1}) \leq \sum_{e \in E_i} f(e \mid B_{i-1}). \tag{13}$$

Consider a single element $e \in E_i$. There are two possible cases: $f(e) < \frac{\tau}{2^{i-1}}$, and $f(e) \geq \frac{\tau}{2^{i-1}}$. In the first case, $f(e \mid B_{i-1}) \leq f(e) < \frac{\tau}{2^{i-1}}$. In the second one, as $|B_{i-1}| < 2^{i-1}$ we conclude $f(e \mid B_{i-1}) < \frac{\tau}{2^{i-1}}$, as otherwise the streaming algorithm would place $e$ in $B_{i-1}$. These observations together with (13) imply:

$$f(E_i \mid B_{i-1}) < \sum_{e \in E_i} \frac{\tau}{2^{i-1}} = \frac{\tau}{2^{i-1}}|E_i|.$$

$\square$

**Lemma B.2** *For every partition $i$, let $B_i$ denote a bucket such that $|B_i| < 2^i$ (i.e. no partition is fully populated), and let $E_i = B_i \cap E$ denote the elements that are removed from $B_i$. The loss in the bucket $B_{\lceil \log k \rceil}$ due to the removals, given all the remaining elements in the previous buckets, is at most*

$$f\left( E_{\lceil \log k \rceil} \ \middle| \ \bigcup_{j=0}^{\lceil \log k \rceil - 1} (B_j \setminus E_j) \right) \leq \sum_{j=1}^{\lceil \log k \rceil} \frac{\tau}{2^{j-1}}|E_j|.$$

*Proof.* We proceed by induction. More precisely, we show that for any $i \geq 1$ the following holds

$$f\left( E_i \ \middle| \ \bigcup_{j=0}^{i-1} (B_j \setminus E_j) \right) \leq \sum_{j=1}^{i} \frac{\tau}{2^{j-1}}|E_j|. \tag{14}$$

Once we show that (14) holds, the lemma will follow immediately by setting $i = \lceil \log k \rceil$.

**Base case $i = 1$.** Since $B_0$ is not fully populated and the maximum number of elements in the partition $i = 0$ is 1, it follows that both $B_0$ and $E_0$ are empty. Then the term on the left hand side of (14) for $i = 1$ becomes $f(E_1)$. As $|B_0| < 1$ we can apply Lemma B.1 to obtain

$$f(E_1) = f(E_1 \mid B_0) \leq |E_1|\frac{\tau}{2^0}.$$

**Inductive step $i > 1$.** Now we show that (14) holds for $i > 1$, assuming that it holds for $i - 1$. First, due to submodularity we have

$$f\left( E_{i-1} \ \middle| \ \bigcup_{j=0}^{i-2} (B_j \setminus E_j) \right) \geq f\left( E_{i-1} \ \middle| \ \bigcup_{j=0}^{i-1} (B_j \setminus E_j) \right),$$

and, hence, we can write

$$f\left( E_i \ \middle| \ \bigcup_{j=0}^{i-1} (B_j \setminus E_j) \right) \leq f\left( E_i \ \middle| \ \bigcup_{j=0}^{i-1} (B_j \setminus E_j) \right) + f\left( E_{i-1} \ \middle| \ \bigcup_{j=0}^{i-2} (B_j \setminus E_j) \right) - f\left( E_{i-1} \ \middle| \ \bigcup_{j=0}^{i-1} (B_j \setminus E_j) \right)$$

$$= f\left( E_i \cup \bigcup_{j=0}^{i-1} (B_j \setminus E_j) \right) + f\left( E_{i-1} \ \middle| \ \bigcup_{j=0}^{i-2} (B_j \setminus E_j) \right) - f\left( E_{i-1} \cup \bigcup_{j=0}^{i-1} (B_j \setminus E_j) \right). \tag{15}$$

Due to monotonicity, the first term can be further bounded by

$$f\left( E_i \cup \bigcup_{j=0}^{i-1} (B_j \setminus E_j) \right) \leq f\left( E_i \cup B_{i-1} \cup \bigcup_{j=0}^{i-2} (B_j \setminus E_j) \right), \tag{16}$$

and for the third term we have

$$f\left(E_{i-1} \cup \bigcup_{j=0}^{i-1} (B_j \setminus E_j)\right) = f\left(E_{i-1} \cup B_{i-1} \cup \bigcup_{j=0}^{i-2} (B_j \setminus E_j)\right) \geq f\left(B_{i-1} \cup \bigcup_{j=0}^{i-2} (B_j \setminus E_j)\right),$$

(17)

where to obtain the identity we used that $E_{i-1} \cup (B_{i-1} \setminus E_{i-1}) = E_{i-1} \cup B_{i-1}$.

By substituting the obtained bounds (16) and (17) in (15) we obtain:

$$f\left(E_i \,\middle|\, \bigcup_{j=0}^{i-1} (B_j \setminus E_j)\right) \leq f\left(E_i \,\middle|\, B_{i-1} \cup \bigcup_{j=0}^{i-2} (B_j \setminus E_j)\right) + f\left(E_{i-1} \,\middle|\, \bigcup_{j=0}^{i-2} (B_j \setminus E_j)\right)$$

$$\leq f\left(E_i \mid B_{i-1}\right) + f\left(E_{i-1} \,\middle|\, \bigcup_{j=0}^{i-2} (B_j \setminus E_j)\right),$$

(18)

where the second inequality follows by submodularity.

Next, Lemma B.1 can be used (as $|B_{i-1}| < 2^{i-1}$) to bound the first term in (18):

$$f\left(E_i \,\middle|\, \bigcup_{j=0}^{i-1} (B_j \setminus E_j)\right) \leq \frac{\tau}{2^{i-1}}|E_i| + f\left(E_{i-1} \,\middle|\, \bigcup_{j=0}^{i-2} (B_j \setminus E_j)\right).$$

(19)

To conclude the proof, we use the inductive hypothesis that (14) holds for $i-1$, which together with (19) implies

$$f\left(E_i \,\middle|\, \bigcup_{j=0}^{i-1} (B_j \setminus E_j)\right) \leq \frac{\tau}{2^{i-1}}|E_i| + \sum_{j=1}^{i-1} \frac{\tau}{2^{j-1}}|E_j| = \sum_{j=1}^{i} \frac{\tau}{2^{j-1}}|E_j|,$$

as desired. □

**Lemma 4.3** *If there does not exist partition of $S$ such that at least half of its buckets are full, then for the set $Z$ produced by* STAR-T-GREEDY *we have*

$$f(Z) \geq \left(1 - e^{-1/3}\right)\left(f\left(B_{\lceil \log k \rceil}\right) - \frac{4m}{wk}\tau\right),$$

*where $B_{\lceil \log k \rceil}$ is a bucket in the last partition which is not fully populated minimizing $\left|B_{\lceil \log k \rceil} \cap E\right|$ and $|E| \leq m$.*

*Proof.* Let $B_i$ denote a bucket in partition $i$ which is not fully populated ($B_i \leq \min\{2^i, k\}$), and for which $|E_i|$, where $E_i = B_i \cap E$, is of minimum cardinality. Such bucket exists in every partition $i$ due to the assumption of the lemma that more than a half of the buckets are not fully populated.

First,

$$f\left(\bigcup_{i=0}^{\lceil \log k \rceil} (B_i \setminus E_i)\right) \geq f\left(B_{\lceil \log k \rceil}\right) - f\left(E_{\lceil \log k \rceil} \,\middle|\, \bigcup_{i=0}^{\lceil \log k \rceil - 1} (B_i \setminus E_i)\right)$$

(20)

$$\geq f\left(B_{\lceil \log k \rceil}\right) - \sum_{i=1}^{\lceil \log k \rceil} \frac{\tau}{2^{i-1}}|E_i|,$$

(21)

where Eq. (20) follows from Lemma D.1 by setting $B = B_{\lceil \log k \rceil}$, $R = E_{\lceil \log k \rceil}$ and $A = \bigcup_{i=0}^{\lceil \log k \rceil - 1}(B_i \setminus E_i)$. As we consider buckets that are not fully populated, Lemma B.2 is used to obtain Eq. (21). Next, we bound each term $\frac{\tau}{2^{i-1}}|E_i|$ in Eq. (21) independently.

From Algorithm 1 we have that partition $i$ consists of $w\lceil k/2^i \rceil$ buckets. By the assumption of the lemma, more than half of those are not fully populated. Recall that $B_i$ is defined to be a bucket of

partition $i$ which is not fully populated and which minimizes $|E_i|$. Let $\tilde{E}_i$ be the subset of $E$ that intersects buckets of partition $i$. Then, $|E_i|$ can be bounded as follows:

$$|E_i| \leq \frac{|\tilde{E}_i|}{\frac{w\lceil k/2^i \rceil}{2}} \leq \frac{2^{i+1}|\tilde{E}_i|}{wk}.$$

Hence, the sum on the left hand side of Eq. (21) can be bounded as

$$\sum_{i=1}^{\lceil \log k \rceil} \frac{\tau}{2^{i-1}}|E_i| \leq \sum_{i=1}^{\lceil \log k \rceil} \frac{\tau}{2^{i-1}} \frac{2^{i+1}|\tilde{E}_i|}{wk} = \frac{4}{wk}\tau \sum_{i=1}^{\lceil \log k \rceil} |\tilde{E}_i| \leq \frac{4|E|}{wk}\tau.$$

Putting the last inequality together with Eq. (21) we obtain

$$f\left(\bigcup_{i=0}^{\lceil \log k \rceil}(B_i \setminus E_i)\right) \geq f\left(B_{\lceil \log k \rceil}\right) - \frac{4|E|}{wk}\tau.$$

Observe also that

$$\bigcup_{i=0}^{\lceil \log k \rceil}|B_i \setminus E_i| \leq \bigcup_{i=0}^{\lceil \log k \rceil}|B_i| \leq k + \bigcup_{i=0}^{\lfloor \log k \rfloor}2^i \leq 3k,$$

which implies

$$f\left(\text{OPT}(3k, S \setminus E)\right) \geq f\left(\bigcup_{i=0}^{\lceil \log k \rceil}(B_i \setminus E_i)\right) \geq f\left(B_{\lceil \log k \rceil}\right) - \frac{4|E|}{wk}\tau.$$

Finally,

$$\begin{aligned}
f(Z) = f(\text{GREEDY}(k, S \setminus E)) &\geq \left(1 - e^{-1/3}\right) f\left(\text{OPT}(3k, S \setminus E)\right) \\
&\geq \left(1 - e^{-1/3}\right)\left(f\left(B_{\lceil \log k \rceil}\right) - \frac{4|E|}{wk}\tau\right) \\
&\geq \left(1 - e^{-1/3}\right)\left(f\left(B_{\lceil \log k \rceil}\right) - \frac{4m}{wk}\tau\right),
\end{aligned} \tag{22}$$

as desired. □

## C  Detailed Proof of Lemma 4.4

**Lemma 4.4** *If there does not exist partition of $S$ such that at least half of its buckets are full, then for the set $Z$ produced by* STAR-T-GREEDY,

$$f(Z) \geq (1 - e^{-1})\big(f(OPT(k, V \setminus E)) - f(B_{\lceil \log k \rceil}) - \tau\big),$$

*where $B_{\lceil \log k \rceil}$ is any bucket in the last partition which is not fully populated.*

*Proof.* Let $B_{\lceil \log k \rceil}$ denote a bucket in the last partition which is not fully populated. Such bucket exists due to the assumption of the lemma that more than a half of the buckets are not fully populated.

Let $X$ and $Y$ be two sets such that $Y$ contains all the elements from $\text{OPT}(k, V \setminus E)$ that are placed in the buckets that precede bucket $B_{\lceil \log k \rceil}$ in $S$, and let $X := \text{OPT}(k, V \setminus E) \setminus Y$. In that case, for every $e \in X$ we have

$$f\left(e \mid B_{\lceil \log k \rceil}\right) < \frac{\tau}{k} \tag{23}$$

due to the fact that $B_{\lceil \log k \rceil}$ is the bucket in the last partition and is not fully populated.

We proceed to bound $f(Y)$:

$$f(Y) \geq f(\text{OPT}(k, V \setminus E)) - f(X) \tag{24}$$

$$\geq f(\text{OPT}(k, V \setminus E)) - f\left(X \mid B_{\lceil \log k \rceil}\right) - f\left(B_{\lceil \log k \rceil}\right) \tag{25}$$

$$\geq f(\text{OPT}(k, V \setminus E)) - f\left(B_{\lceil \log k \rceil}\right) - \sum_{e \in X} f\left(e \mid B_{\lceil \log k \rceil}\right) \tag{26}$$

$$\geq f(\text{OPT}(k, V \setminus E)) - f\left(B_{\lceil \log k \rceil}\right) - \frac{\tau}{k}|X| \tag{27}$$

$$\geq f(\text{OPT}(k, V \setminus E)) - f\left(B_{\lceil \log k \rceil}\right) - \tau, \tag{28}$$

where Eq. (24) follows from $f(\text{OPT}(k, V \setminus E)) = f(X \cup Y)$ and submodularity, Eq (25) and Eq (26) follow from monotonicity and submodularity, respectively. Eq. (27) follows from Eq. (23), and Eq. (28) follows from $|X| \leq k$.

Finally, we have:

$$f(Z) = f(\text{GREEDY}(k, S \setminus E)) \geq \left(1 - e^{-1}\right) f(\text{OPT}(k, S \setminus E))$$

$$\geq \left(1 - e^{-1}\right) f(\text{OPT}(k, Y)) \tag{29}$$

$$= \left(1 - e^{-1}\right) f(Y) \tag{30}$$

$$\geq \left(1 - e^{-1}\right) \left(f(\text{OPT}(k, V \setminus E)) - f(B_{\lceil \log k \rceil}) - \tau\right), \tag{31}$$

where Eq. (29) follows from $Y \subseteq (S \setminus E)$, Eq. (30) follows from $|Y| \leq k$, and Eq. (31) follows from Eq. (28). □

# D  Technical Lemma

Here, we outline a technical lemma that is used in the proof of Lemma 4.3

**Lemma D.1** *For any submodular function $f$ on a ground set $V$, and any sets $A, B, R \subseteq V$, we have*

$$f(A \cup B) - f(A \cup (B \setminus R)) \leq f\left(R \mid A\right).$$

*Proof.* Define $R_2 := A \cap R$, and $R_1 := R \setminus A = R \setminus R_2$. We have

$$f(A \cup B) - f(A \cup (B \setminus R)) = f(A \cup B) - f((A \cup B) \setminus R_1)$$

$$= f\left(R_1 \mid (A \cup B) \setminus R_1\right)$$

$$\leq f\left(R_1 \mid (A \setminus R_1)\right) \tag{32}$$

$$= f\left(R_1 \mid A\right) \tag{33}$$

$$= f\left(R_1 \cup R_2 \mid A\right) \tag{34}$$

$$= f\left(R \mid A\right),$$

where (32) follows from the submodularity of $f$, (33) follows since $A$ and $R_1$ are disjoint, and (34) follows since $R_2 \subseteq A$. □

# E  Detailed Proof of Theorem 4.5

Setting $\tau$ in STAR-T assumes that we know the unknown value $f(\text{OPT}(k, V \setminus E))$. In this subsection we show how to approximate that value. First, $f(\text{OPT}(k, V \setminus E))$ can be bounded in the following way: $\eta \leq f(\text{OPT}(k, V \setminus E)) \leq k\eta$, where $\eta$ denotes the largest value of any of the elements of $V \setminus E$, i.e. $\eta = \max_{e \in (V \setminus E)} f(e)$. In case we are given $\eta$, we follow the same approach as in [8] by considering all the $O\left(\log_{1+\epsilon} k\right)$ possible values of $f(\text{OPT}(k, V \setminus E))$ from the set $\{(1+\epsilon)^i \mid i \in \mathbb{Z}, \eta \leq (1+\epsilon)^i \leq k\eta\}$. For each of the thresholds independently and in parallel we then run STAR-T, and hence build $O\left(\log_{1+\epsilon} k\right)$ different summaries. After the stream ends, on each of the summaries we run algorithm STAR-T-GREEDY and report the maximum output over all the runs.

---

**Algorithm 3** Parallel Instances of (STAR-T)

**Input:** Set $V, k, w \in \mathbb{N}_+, \eta \in \mathbb{R}$
1: $O = \left\{ (1 + \epsilon)^i \mid \eta \leq (1 + \epsilon)^i \leq k\eta \right\}$
2: Create a set of instances $\mathcal{I} := \{\text{STAR-T}(V, k, \eta, w) \mid \eta \in O\}$, and run all the instances in parallel over the stream.
3: Let $\mathcal{S} = \{\text{the output of instance } I \mid I \in \mathcal{I} \}$.
4: **return** $\mathcal{S}$

---

**Algorithm 4** Parallel Instances STAR-T- GREEDY

**Input:** Family of sets $\mathcal{S}$, query set $E$ and $k$
1: $Z \leftarrow \arg\max_{S \in \mathcal{S}} \text{GREEDY}(k, S \setminus E)$
2: **return** $Z$

---

As this approach runs $O(\log_{1+\epsilon} k)$ copies of our algorithm, it requires $O(\log_{1+\epsilon} k)$ more memory space than stated in Theorem 4.1. Furthermore, since we are approximating $f(\text{OPT}(k, V \setminus E))$ as the geometric series with base $(1 + \epsilon)$, our final result is an $(1 + \epsilon)$-approximation of the value provided in the theorem.

Unfortunately, the value $\eta$ might also not be known a priori. However, $\eta$ is some value among the $m + 1$ largest elements of the stream. This motivates the following idea. At every moment, we keep $m + 1$ largest elements of the stream. Let $L$ denote that set (note that $L$ changes during the course of the stream). Then, for different values of $\eta$ belonging to the set $\{f(e) \mid e \in L\}$ we approximate $f(\text{OPT}(k, V \setminus E))$ as described above. Here we make a minor difference, as also described in [8]. Namely, instead of instantiating all the copies of the algorithm corresponding to $\eta \leq (1 + \epsilon)^i \leq km$, we instantiate copies of the algorithm corresponding to the values of $f(\text{OPT}(k, V \setminus E))$ from the set $\{(1 + \epsilon)^i \mid i \in \mathbb{Z}, \eta \leq (1 + \epsilon)^i \leq 2k\eta\}$. We do so as an element $e$ can belong to an instance of our algorithm even if $f(\text{OPT}(k, V \setminus E)) = 2kf(e)$.

Next, let $e$ be a new element that arrives on the stream. If $e$ is not among the $m + 1$ largest elements of the stream seen so far, we do not instantiate any new copy of our algorithm. On the other hand, if $e$ should replace another element $e' \in L$ because $e'$ does not belong to the $m + 1$ largest elements of the stream anymore, we redefine $L$ to be $(L \setminus \{e'\}) \cup \{e\}$, and update the instances. The instances are updated as follows: we instantiate copies (those that do not exist already) of our algorithm for $\eta = f(e)$ as described above; and, any instance of our algorithm corresponding to $\eta = f(e')$, but not to any other element of $L$, we discard.

To bound the space complexity, we start with the following observation – given an element $e$, we do not need to add $e$ to any instance of our algorithm corresponding to $f(\text{OPT}(k, V \setminus E)) < f(e)$. This reasoning is justified by the following: if $e \in E$, then it does not matter whether we keep $e$ in our summary or not; if $e \notin E$, then $f(\text{OPT}(k, V \setminus E)) \geq f(e)$. Therefore, those thresholds that are less than $f(e)$ are not a good estimate of the optimum solution with respect to $e$. To keep the memory space low, we pass an element $e$ to the instances of our algorithm corresponding to the of $f(\text{OPT}(k, V \setminus E))$ being in set $\{(1 + \epsilon)^i \mid i \in \mathbb{Z}, f(e) \leq (1 + \epsilon)^i \leq 2kf(e)\}$. Notice that, by the structure of our algorithm, $e$ will not be added to any instance of our algorithm with threshold more than $2kf(e)$.

Putting all together we make the following conclusions. At any point during the execution, every element of $L$ belongs to at most $O(\log_{1+\epsilon} k)$ instances of our algorithm. Define $e_{\min} := \arg\min_{e \in L} f(e)$. Then by the definition, every element $a \notin L$ kept in the parallel instances of our algorithms is such that $f(a) \leq f(e_{\min})$. This further implies that $a$ also belongs to at most $O(\log_{1+\epsilon} k)$ instances corresponding to the following set of values $\{(1 + \epsilon)^i \mid i \in \mathbb{Z}, f(e_{\min}) \leq (1 + \epsilon)^i \leq 2kf(e_{\min})\}$. Therefore, the total memory usage of the elements of $L$ is $O\left(m \log_{1+\epsilon} k\right)$. On the other hand, since all the elements not in $L$ belong to at most $O(\log_{1+\epsilon} k)$ different instances of STAR-T, the total memory those elements occupy is $O((k + m \log k) \log k \log_{1+\epsilon} k)$. Therefore, the memory complexity of this approach is $O\left((k + m \log k) \log k \log_{1+\epsilon} k\right)$

# F   Additional results for the dominating set problem

In Figure 3 we outline further results for the dominating set problem considered in Section 5.1.

Figure 3: Numerical comparisons of the algorithms STAR-T-GREEDY, STAR-T-SIEVE and SIEVE-STREAMING.