[Reviews · NeurIPS 2017]

Reviewer 1



Problem studied : The paper studies the following question. The goal is to find a set of elements T from some ground set E in a streaming fashion such that, if we delete any set of m elements from T then we can still find a set of k elements S such that f(S) is maximized where f is a submodular function. Applications: Maximizing submodular functions has multiple applications and we know streaming algorithms for this problem. The reason why a robust version is useful is if we find a solution and then we might need to remove some of the elements (say if you compute a set of movies for the user and then find that the user is not interested in some of them). Results: The paper gives a constant factor approximation to this problem while returning a set T of size (m*logk+k)*log^2k. Comments: The paper's results are correct to the best of my understanding. The solution is non-trivial and the algorithm and it's analysis are technically challenging. While the problem is interesting my main concern is the relevance of this paper to the conference. The applications and their motivation are quite weak and the problem is not general enough.

Reviewer 2



This paper proposes a two-stage procedure for robust submodular maximization with cardinality constraint k. In the first stage, a robust streaming algorithm STAR-T is designed, which is based on a partitioning structure and an decreasing thresholding rule. In the second stage, the simple greedy method is used. The STAR-T is proven to have constant approximation guarantee when k is up to infinity (Thm4.1). lem4.2~4.3 prove that STAR-T with simple greedy method can deal with the loss of elements. Experimental results show that the proposed algorithm can do well on the problem of influence maximization and personalized movie recommendation. This work is interesting as it is the first streaming algorithm for robust submodular maximization. Maybe the streaming setting will have wide range of applications and this work is very enlightening. The STAR-T algorithm has two hyper parameters, \tao and w. In Thm 4.5, the authors showed how to approximate \tao. The authors showed “w=1” can do well in Experiments. It’s convenient to use STAR-T without trying different hyper parameters. The major problem I concerned is that the experiment is not very expressive. The STAR-T-Greedy algorithm can not always outperform the Sieve-St algorithm. In addition, it’s strange that the objective function of Influence Maximization is f(Z)=|N(Z) \cup Z| (line 219 in this paper), which is not the same with the function in the referred article (D. Kempe, J. Kleinberg, and É. Tardos, “Maximizing the spread of influence through a social network,” in Int. Conf. on Knowledge Discovery and Data Mining (SIGKDD), 2003). It is not clear to me that how STAR-T performs on such a spread model.

Reviewer 3



This paper studies the problem of submodular maximization in a streaming and robust setting. In this setting, the goal is to find k elements of high value among a small number of elements retained during the stream, where m items might be removed after the stream. Streaming submodular maximization has been extensively studied due to the large scale applications where only a small number of passes over the data is reasonable. Robustness has also attracted some attention recently and is motivated by scenarios where some elements might get removed and the goal is to obtain solutions that are robust to such removals. This paper is the first to consider jointly the problem of streaming and robust submodular maximization. The authors develop the STAR-T algorithm, which maintains O((m log k + k )log^2 k) elements after the stream and achieves a constant (0.149) approximation guarantee while being robust to the removal of m elements. This algorithm combines partitioning and thresholding techniques for submodular optimization which uses buckets. Similar ideas using buckets were used in previous work for robust submodular optimization, but here the buckets are filled in a different fashion. It is then shown experimentally that this algorithm performs well in influence maximization and movie recommendation applications, even against streaming algorithms which know beforehand which elements are removed. With both the streaming and robustness aspects, the problem setting is challenging and obtaining a constant factor approximation is a strong result. However, there are several aspects of the paper that could be improved. It would have been nice if there were some non-trivial lower bounds to complement the upper bound. The writing of the analysis could also be improved, the authors mention an “innovative analysis” but do not expand on what is innovative about it and it is difficult to extract the main ideas and steps of the analysis from the proof sketch. Finally, the influence maximization application could also be more convincing. For the experiments, the authors use a very simplified objective for influence maximization which only consists of the size of the neighborhood of the seed nodes. It is not clear how traditional influence maximization objectives where the value of the function depends on the entire network for any set of seed nodes could fit in a streaming setting.